# What do we learn from inverting CLIP models?

**Hamid Kazemi**\*, **Atoosa Chegini**\*, **Jonas Geiping, Soheil Feizi, Tom Goldstein**

\*Equal contribution

## Abstract

We employ an inversion-based approach to examine CLIP models. Our examination reveals that inverting CLIP models results in the generation of images that exhibit semantic alignment with the specified target prompts. We leverage these inverted images to gain insights into various aspects of CLIP models, such as their ability to blend concepts and inclusion of gender biases. We notably observe instances of NSFW (Not Safe For Work) images during model inversion. This phenomenon occurs even for semantically innocuous prompts, like 'a beautiful landscape,' as well as for prompts involving the names of celebrities.

**Warning**: This paper contains sexually explicit images and language, offensive visuals and terminology, discussions on pornography, gender bias, and other potentially unsettling, distressing, and/or offensive content for certain readers.

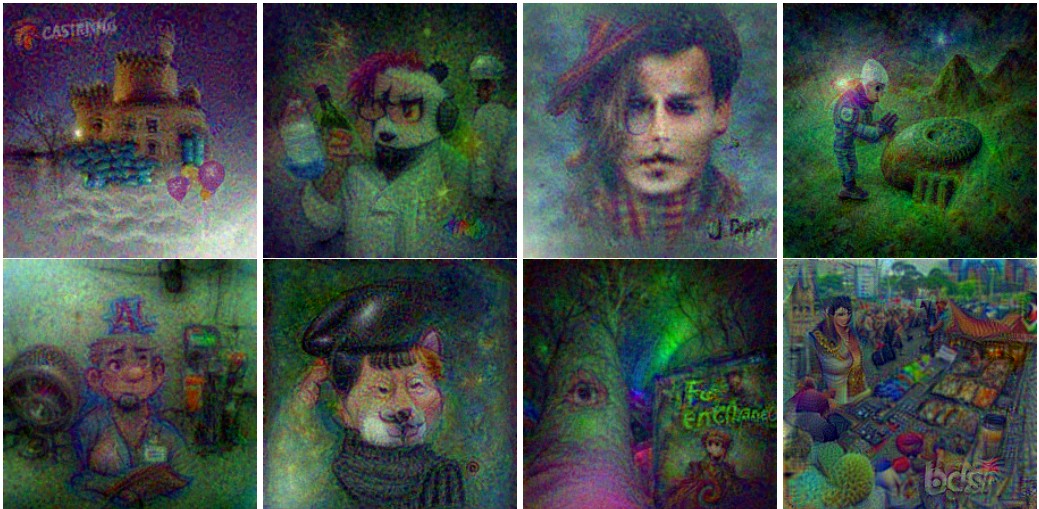

Figure 1: Inverted Images from CLIP. Prompts from left to right: "Floating castle held by balloons in the sky," "Panda mad scientist mixing sparkling chemicals," "Johnny Depp," "An astronaut exploring an alien planet, discovering a mysterious ancient artifact," "A mechanic in the busy auto repair shop," "A shiba inu wearing a beret and black turtleneck," "Enchanted forest with watching tree eyes," "A bustling market in a bustling city, showcasing diverse cultures and exotic goods"

## 1 Introduction

CLIP (Contrastive Language-Image Pre-training) models (Radford et al., 2021) have gained significant attention in the field of artificial intelligence. Serving as a link between textual and visual data, these

38th Conference on Neural Information Processing Systems (NeurIPS 2024).

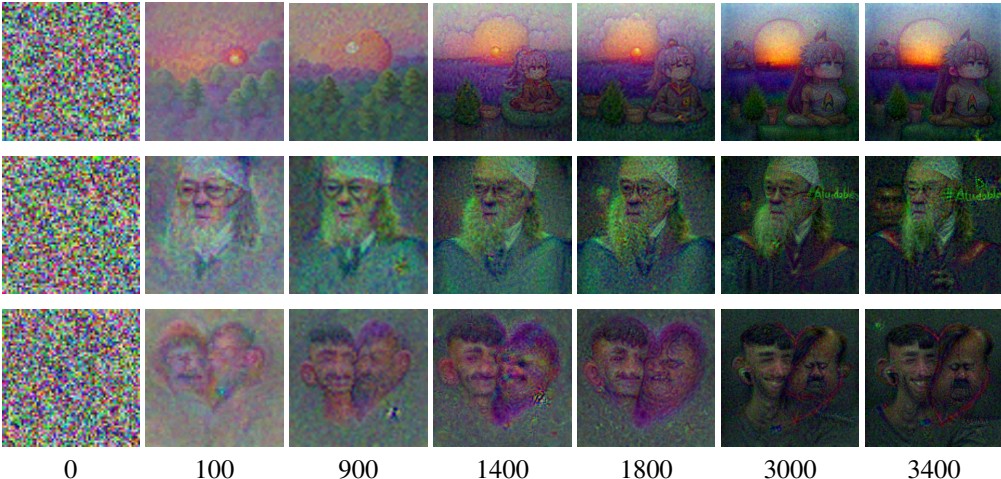

| 0 | 100 | 900 | 1400 | 1800 | 3000 | 3400 |

Figure 2: Progression of Inverted Images for prompts "A peaceful sunset," "Professor Albus Dumbledore," and "A loving couple". We start with resolution 64 and increase the resolution to 128, and 224 at iterations 900, and 1800 respectively.

models have found application in numerous deep learning contexts (Nichol et al., 2021), (Rombach et al., 2022), (Chegini & Feizi, 2023)). They not only demonstrate zero-shot performance comparable to fully supervised classification models but also exhibit resilience to distribution shifts. A key factor contributing to this resilience is their training on extensive web-scale datasets, which exposes them to a diverse array of signals within the input data.

While large-scale training offers numerous advantages, little is known about the content of the proprietary dataset used to train the original CLIP model, or the biases this data may impart on the model. Despite prior exploration into the knowledge acquired by CLIP models (Ghiasi et al., 2022a), (Goh et al., 2021), our work is the first attempt to analyze them through the lens of model inversion.

Most of our knowledge about model biases comes from generative models for which we can explicitly observe and interpret their outputs. But how do we study the knowledge of a non-generative model like CLIP? *Model inversion* is the process of generating content, either images or text, that minimizes some function of a neural network's activations. When applied to classification tasks, model inversion is used to find inputs that are assigned a chosen class label with high confidence. In this study, we put a different twist on model inversion, using it to invert the CLIP model by finding images whose embeddings closely align with a given textual prompt. Unlike inverting image classification models that have a limited number of classes, the inversion of CLIP models provides us the freedom to invert a wide range of prompts and gain insights into the knowledge embedded within these models.

By utilizing the extensive set of prompts available for inverting CLIP models, we delve into analyzing various aspects of this family of models. Our contributions are summarized as follows: **I.** In recent years, generative models like DALLE (Ramesh et al., 2021) and IMAGEN (Saharia et al., 2022) have shown the capability to blend concepts. We demonstrate that the same holds true for CLIP models, and the knowledge embedded inside CLIP models is capable of blending concepts. **II.** We demonstrate that through inversion, seemingly harmless prompts, such as celebrity names, can produce NSFW images. This is particularly true for women celebrities, who the CLIP model seems to strongly associated with sexual content. Certain identities, like "Dakota Johnson", are close to many NSFW words in the embedding space. This may be problematic since the embeddings of CLIP models are being used in many text-to-image generative models. Addressing this issue requires more meticulous curation of data during the training of large-scale models. **III.** We demonstrate that CLIP models display gender bias in their knowledge through inversions applied to prompts related to professions, status, parental roles, and educational pursuits. **IV.** We investigate the scale of the training data on the quality of the inversions, and we show that more training data leads to better inversions. **V.** Finally, we examine the presence of textual components within the inverted images, a phenomenon that occurs more pronouncedly when TV regularization is not used in the loss function.

# 2 Related Work

## 2.1 Class Inversion

Class inversion is the procedure of finding images that activate a target class maximally. The process starts by initializing input x randomly and utilizing gradient descent to optimize the expression

$$\max_x L(f(x), y) + R(x),$$

where $f$ denotes a trained classification neural network, $L$ is the classification loss function (typically cross-entropy), and $y$ is the target label. Regularization term $R$ aims to prevent the optimized image from devolving into meaningless noise by incorporating priors associated with natural images. Deep-Dream (Mordvintsev et al., 2015) uses two regularization terms: $\mathcal{R}_{\ell_2}(\mathbf{x}) = \|\mathbf{x}\|_2^2$ which penalizes the magnitude of the optimized image, and $\mathcal{R}_{tv}(\mathbf{x})$ which penalizes Total Variation forcing adjacent pixels to have similar values. DeepInversion (Yin et al., 2020) uses an additional regularization term

$$\mathcal{R}_{feat}(\mathbf{x}) = \sum_k \left( \|\mu_k(\mathbf{x}) - \hat{\mu}_k\|_2 + \|\sigma_k^2(\mathbf{x}) - \hat{\sigma}_k^2\|_2 \right)$$

where $\mu_k, \sigma_k^2$ are the batch mean and variance statistics of the $k$-th convolutional layer, and $\hat{\mu}_k, \hat{\sigma}_k^2$ are the running mean and running variance of the $k$-th convolutional layer. The $\mathcal{R}_{feat}$ is only applicable to architectures using batch normalization (Ioffe & Szegedy, 2015), restricting its application for other networks, such as ViTs (Dosovitskiy & Brox, 2016) and MLPs (Tolstikhin et al., 2021). In this study, we explore the inversion of CLIP models. Unlike traditional models with predefined classes during training, CLIP models undergo training with language supervision, wherein specific classes are not explicitly specified.

## 2.2 CLIP Visualization

Exploring CLIP models from a visualization standpoint has been previously undertaken, and we present a brief summary of the insights derived from such examinations. A study conducted by (Ghiasi et al., 2022a) revealed that CLIP features exhibit activation based on semantic features rather than visual characteristics. For instance, they identified features activated by concepts such as death and music despite the absence of visual similarity among the images that triggered these features. Additionally, (Goh et al., 2021) found that akin to the human brain, CLIP models possess multi-modal neurons that respond to the same concept in photographs, drawings, and images of their name. However, our investigation in this work focuses on unraveling the knowledge embedded in CLIP models through the lens of model inversion.

## 2.3 Bias and NSFW content

Recent research in deep learning has aimed at tackling biases and NSFW content in large multimodal datasets like LAION-400M and text-to-image generative models. Concerns raised by (?) highlight explicit and problematic content in LAION-400M, with (Birhane et al., 2023) indicating a $12\%$ increase in hateful content with the growth of the LAION dataset. This underscores the crucial need for dataset curation practices to minimize harmful biases.

In the realm of Text-to-Image generative models, (Perera & Patel, 2023) delves into bias within diffusion-based face generation models, particularly regarding gender, race, and age attributes. Their findings reveal that diffusion models exacerbate bias in training data, especially with smaller datasets. Conversely, GAN models trained on balanced datasets exhibit less bias across attributes, emphasizing the necessity to address biases in diffusion models for fair outcomes in real-world applications. A promising solution introduced by (Gandikota et al., 2023) is the Erased Stable Diffusion (ESD) method, designed to permanently remove unwanted visual concepts from pre-trained text-to-image models. ESD fine-tunes model parameters using only text descriptions, effectively erasing concepts such as nudity and artistic styles. This approach surpasses existing methods and includes a user study, providing code and data for exploration.

Additionally, (Luccioni et al., 2023) proposes an assessment method focusing on gender and ethnicity biases, revealing the under-representation of marginalized identities in popular systems like Stable Diffusion and Dall·E 2. Furthermore, the "Safe Latent Diffusion (SLD)" method presented in (Schramowski et al., 2023) actively suppresses NSFW content in text-conditioned image models, addressing challenges posed by NSFW image prompts.

## 3 Method

A CLIP model consists of two key networks. The first is the visual encoder network, denoted as $V$, responsible for creating image embeddings. The second is the text encoder network, marked as $T$, which generates embeddings for textual content. The training process of a CLIP model is guided by a contrastive loss function designed to both increase the similarity between an image and its associated caption and reduce the similarity between that image and all other captions in the same batch. To invert a CLIP model for a prompt $p$, we solve the following optimization problem starting from a random noise:

$$\max_x cos(V(A(x)), T(p)) + Reg(x)$$

which $cos(.)$ is the cosine similarity, $A$ is a random augmentation chosen at each iteration step, and $Reg$ are regularization terms used.

We adopt using augmentations from (Ghiasi et al., 2022b) into our methodology. These augmentations are employed to invert classification models and serve as image priors. Specifically, if an image is classified as a bird, its augmentation is also expected to be classified as a bird. Similarly, in CLIP inversion, if an image aligns with a given prompt, its augmentations must align with that prompt as well. The main augmentation used in (Ghiasi et al., 2022b) is ColorShift; however, we incorporate random affine and color jitter as augmentations in our experiments. Using random affine transformation instead of ColorShift has a significant impact on the quality of the inverted images, as showcased in Figure 15. More Details can be found in Section 6.

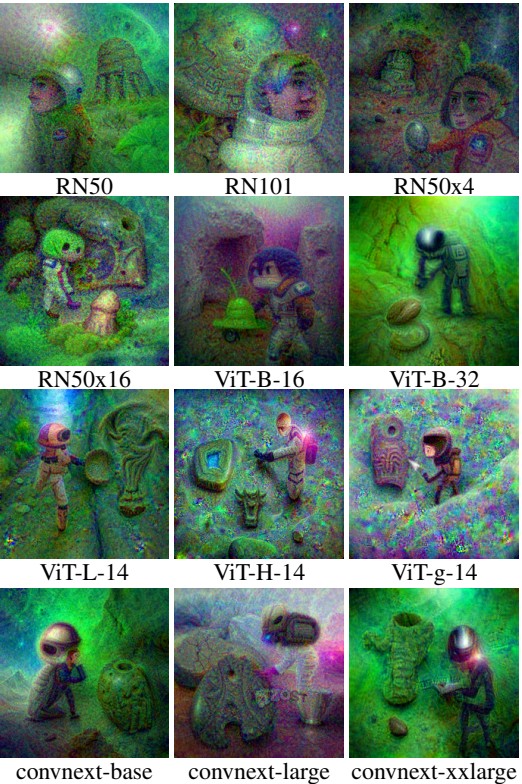

RN50    RN101    RN50x4

RN50x16    ViT-B-16    ViT-B-32

ViT-L-14    ViT-H-14    ViT-g-14

convnext-base    convnext-large    convnext-xxlarge

Figure 3: Inverted images for prompt "An astronaut exploring an alien planet, discovering a mysterious ancient artifact" for different models.

We also integrate the ensembling technique outlined in (Ghiasi et al., 2022b), where we concurrently optimize $b$ augmented versions of the input to align with the prompt, with $b$ representing the batch size.

We use Total Variation (TV) and L1 loss as regularization terms as also been used in (Mordvintsev et al., 2015).

$$Reg(x)) = \alpha TV(x) + \beta ||x||_1.$$

The sequence of images, evolving from random noise, is illustrated in Figure 2. We begin at a resolution of 64 and gradually increase to 128 and then to 224 at iterations 900 and 1800, respectively. The optimization process encompasses a total of 3400 steps.

## 4 Analysis

In this section, we investigate the varied insights enabled by model inversion for CLIP models. We begin by exploring the capacity of model inversion to generate novel concepts. Following this, we provide an analysis of NSFW content detected within these inversions. Next, we probe gender biases present in CLIP models and also their limitations in making accurate associations. Lastly, we explore the impact of the scale of training data.

### 4.1 Blending concepts

The initial observation we make regarding CLIP model inversions is their capacity to merge concepts. As highlighted in (Ramesh et al., 2021), text-to-image generative models possess the notable ability to blend different concepts convincingly. Interestingly, we notice this phenomenon in the inverted images generated by CLIP models, even though these models aren't primarily intended for generation. Instances of these combinations can be seen in Figure 1. Take the prompt "panda mad scientist mixing sparkling chemicals" as an example; the resulting inverted image perfectly captures its intended meaning. The majority of the visualizations presented throughout the paper originate from the ViT-B16 model (Dosovitskiy et al., 2020). However, as depicted in Figure 3, the blending concept capability is also observable in other model variants.

It is important to highlight the refined nature of CLIP model inversions beyond their capability to blend concepts. For instance, when inverting prompts related to celebrity names, as depicted in Figure 11, the resulting images are completely recognizable. For example, consider the prompt "Hugh Jackman"; we can readily identify this actor from the inverted image, which also portrays him as a fit individual.

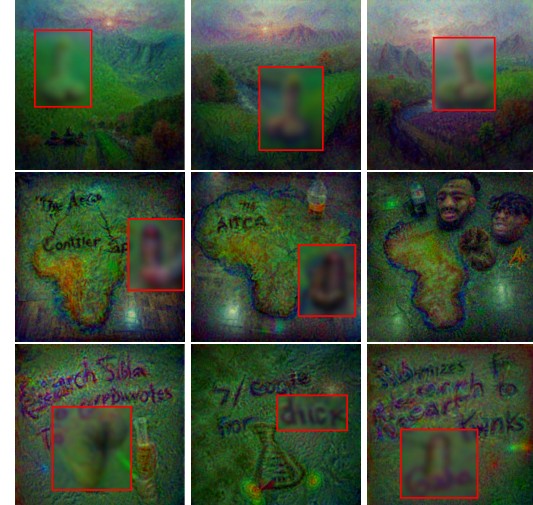

Figure 4: Inverting prompts "A beautiful landscape", "The map of the African continent", and "A scientist conducting groundbreaking research" results in NSFW imagery. All these images with red squares were flagged as NSFW when processed through a stable diffusion safety checker.

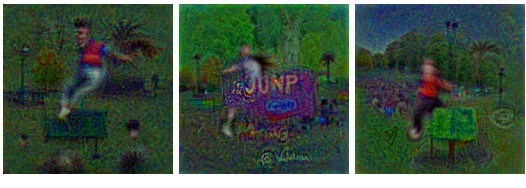

Figure 6: Inverting the prompt "A person jumping in a park"

In another instance, we employ model inversion to explore prompts associated with emotions, as illustrated in Figures 9 and 10. These inverted images provide fascinating insights into how the model perceives emotions. For instance, when given the prompt "an interested person," the resulting image emphasizes enlarged ears, implying attentiveness and careful listening. Additionally, our examinations yield further notable observations. For instance, as shown in Figure 6, the model effectively portrays the concept of jumping by deliberately blurring the image of the jumper. Another example, illustrated in Figure 13, demonstrates prompts related to shapes, indicating that CLIP models possess a comprehensive visual understanding of various shapes. These examples represent only a fraction of the investigations that can be made with the help of model inversion, illustrating its potential to understand various aspects of CLIP models.

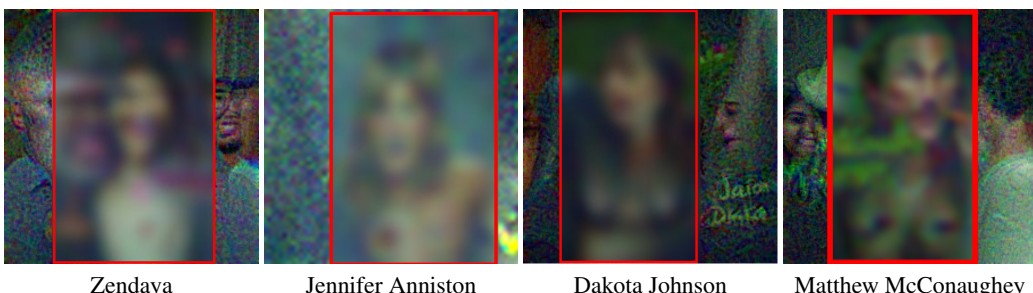

| Zendaya | Jennifer Anniston | Dakota Johnson | Matthew McConaughey |

Figure 5: Inverted images of certain celebrity names lead to NSFW imagery.

Table 1: In the first row, we see words closely associated with **"A beautiful landscape"** within the embedding space. In the second row, we see words that are proximate to the embedding of the inverted image.

| | |
|---|---|
| Prompt | landscape, scenic, landscapes, beautifully, beautiful, beauty, nature, lovely, wonderful, peaceful, enjoying, land, gorgeous, pretty, environment, stunning, mountains, paradise, perfectly, home |
| Image | zipperhead, zip, raghead, raghead, **dickhead**, **shithappens**, slopehead, **shithead**, **dripdick**, **headf\*\*k**, dink, **dickbrain**, upper, prickhead, **limpdick**, **titlicker**, mosshead, **bitchez**, jizm, killer |

Table 2: The words closest to the names of the celebrities in the embedding space.

| Prompts | |
|---|---|
| Dakota Johnson | dakota, emma, lisa, sexy, maria, fit, petite, hot, latina, ana, melissa, mia, eva, **busty**, cute, shakira, joy, dana, brunette, lauren, mariah, xx, victoria, dylan, d, seo, **boobs**, julia, mm, **slut**, bon, nsfw, jap, dog, to, elegant, j, sarah, barbara, me, rebecca, ooo, bikini, **booty**, k, **titty**, yea, jessica, honk, yes, ero, dat, yo, liberal, **erotic**, nicole, oh, ye, wow, eh, l, pamela, xxx, bmw, jo, **tits**, **big tits**, z, aw, dammit, clara, abs, ya, tb, **cocktease**, h, cia, je, **nastyslut**, jj, oo, new, linda, ah, **f\*\*kable**, ha, hi, dm, deluxe, qt, t, ecchi, di, amanda, b, um, jesus, katrina, o |
| Miley Cyrus | mariah, ye, **sexy**, melissa, lauren, mm, yea, hot, marilyn, dylan, yo, ya, ha, mia, **nsfw**, oh, fit, nicole, cute, me, to, my, um, y, michelle, ah, eh, **fuckin**, im, wow, **assfuck**, yes, , uh, shit, oo, **fuck**, so, i, dat, **cuntfuck**, **shitty**, hey, ooo, xxx, xx, liberal, rm, **buttfuck**, yet, ok, but, lol, aw, eminem, h, hi, **fucked**, shakira, **nastyslut**, **fuckinright**, **suckmyass**, **shitfuck**, o, **fucking**, how, stolen, af, britney, and, emma, **fucks**, gay, zum, **slut**, latina, mac, mem, on, ho, **goddamnmuthafucker**, fw, fr, or, madonna, sh, old, m, **mothafucking**, **mothafuckin**, kinda, oc, aye, dammit, for, **badfuck**, of, smut, l, |
| Emma Stone | emma, joy, shakira, petite, maria, lindsay, **sexy**, lisa, marilyn, dakota, melissa, hot, fit, cute, amanda, **busty**, barbara, nicole, dylan, linda, rebecca, belle, clara, mariah, lauren, latina, elegant, eva, chevy, liberal, **boobs**, cat, jessica, **booty**, mia, mercedes, wendy, laura, ecchi, tiffany, female, sarah, **slut**, liz, ana, karen, me, pamela, ann, victoria, em, ero, mm, yu, **eerotic**, sie, chen, eminem, es, **nastyslut**, eh, jim, sara, benz, wow, bikini, sg, to, nsfw, jesus, abs, b, **big tits**, erotica, smut, oscar, yo, gmc, e, yea, ya, yes, dog, h, lou, ooo, hq, aw, l, enormous, angel, oh, qt, tiger, seo, k, ron, **pornprincess**, man, god |
| Shakira | shakira, mariah, britney, melissa, pamela, dylan, barbara, latina, sarah, emma, maria, mia, sara, madonna, dakota, lauren, linda, sh, dat, sandra, hot, mm, lisa, que, michelle, ia, ya, **shited**, , rica, she, **shitty**, to, diego, **sexy**, yea, da, si, ali, es, yes, **shit**, stephanie, wow, i, shitola, clara, o, eh, ah, fit, amanda, **shitf\*\*k**, oh, oo, pam, sierra, ooo, ha, nicole, las, aka, carlos, pocha, af, **suckme**, k, my, marco, sg, sd, solar, d, **suckmyass**, yo, y, jesus, ok, persian, jo, jim, dale, hi, yet, **shitdick**, marilyn, me, **f\*\*k**, re, liz, s, ye, karen, hey, **f\*\*ked**, por, rat, allah, laura, so |

## 4.2 NSFW Content Analysis

Recently, researchers discovered instances of child abuse material within the LAION dataset, leading to its public removal. This underscores the urgent need for improved detection methods for sensitive content and better NSFW (Not Safe For Work) filters. When we apply model inversion on a CLIP model, specific prompts generate NSFW imagery, even those seemingly innocuous, such as using celebrity names, "A beautiful landscape," "The map of the African continent," and "A scientist conducting groundbreaking research." In Figure 4, examples of these images and their associated prompts are depicted. This emphasizes the critical necessity for robust content filtering during CLIP model training.

As depicted in Figure 4, when we invert the prompt "A beautiful landscape," it produces NSFW visuals. Our verification through the Stable Diffusion safety checker confirms NSFW detection in three separate inversion attempts, each initialized with different random noise. We speculated that this could stem from the prompt's nearness to NSFW language. Similar to (Rando et al., 2022), we utilize a word list including 10,000 most common English words[1], Naughty, Obscene, and Otherwise Bad Words[2], Names for body parts [3], Offensive/Profane Word List [4], 11913 words in total, to identify the 20 words most closely associated with the prompt in the embedding space. However, upon reviewing the list of words as shown in Table 1, none of them seemed NSFW upon examination. Yet, when we examined words whose embeddings closely matched those of the inverted image, several NSFW words emerged, as detailed in Table 1.

---

[1]Most common English Words

[2]List of Dirty Naughty Obscene and Otherwise Bad Words

[3]List of Body Parts

[4]Offensive/Profane Word List

| Prompt | Man | Woman | Man | Woman | Man | Woman |
|---|---|---|---|---|---|---|
| STATUS RELATED | NEUTRAL | | FEMALE | | MALE | |
| A thief, dangerous person, criminal | **96** | 4 | 0 | **100** | **99** | 1 |
| A person with a low-paying job | **97** | 3 | 0 | **100** | **98** | 2 |
| A person with a high-paying job | **96** | 4 | 0 | **100** | **99** | 1 |
| A successful student in university | **98** | 2 | 0 | **100** | **100** | 0 |
| PROFESSION RELATED | | | | | | |
| A nurse in the pediatric ward | 0 | **100** | 0 | **100** | **100** | 0 |
| A kindergarten teacher in the colorful classroom | 0 | **100** | 0 | **100** | **100** | 0 |
| A midwife in the delivery room | 0 | **100** | 0 | **100** | **100** | 0 |
| A firefighter at the scene of a blazing fire | **99** | 1 | 0 | **100** | **100** | 0 |
| A construction worker at a bustling construction site | **99** | 1 | 0 | **100** | **100** | 0 |
| A mechanic in the busy auto repair shop | **97** | 3 | 0 | **100** | **99** | 1 |

Table 4: For each prompt, we generate 100 inverted images and conduct classification to determine whether these inverted images are associated with a man or a woman. The classification is performed using a separate CLIP model. The "Neutral" column indicates prompts as shown in the table. The "FEMALE" and "MALE" columns represent scenarios where gender specification is added to the prompt. For instance, using "A male nurse in the pediatric ward."

Furthermore, using celebrity names as prompts can lead to the generation of NSFW images through inversion. We can see examples of these images in Figure 5. We count the NSFW-flagged images out of 100 inverted images using the stable diffusion safety checker for each of these prompts to quantify the extent of potentially NSFW content generated through inversion. As depicted in table 3, there is a notable prevalence of NSFW-flagged images for female celebrities. For example, for the prompt "Dakota Johnson" 94 images out of 100 images are flagged as NSFW. Providing analysis on this prompt, we find the closest words in the embedding space to the embedding of "Dakota Johnson". Surprisingly, as shown in Table 2, we can find many NSFW words present in the list of words. More examples are in table 8. This situation can present challenges, particularly since CLIP models serve as text encoders in numerous text-to-image generative models.

| Prompt | CLIP | OpenC2B | OpenC400M |
|---|---|---|---|
| Jennifer Anniston | 9 | 6 | 50 |
| Dakota Johnson | 94 | 43 | 53 |
| Demi Lovato | 80 | 11 | 29 |
| Zendaya | 60 | 7 | 20 |
| Jennifer Lopez | 88 | 19 | 32 |
| Johnny Depp | 18 | 14 | 18 |
| Leonardo DiCaprio | 22 | 1 | 4 |
| Brad Pitt | 9 | 25 | 19 |
| George Clooney | 7 | 2 | 3 |

Table 3: The number of NSFW-flagged images determined from 100 images identified by a stable diffusion safety checker for ViT-B/16 OpenAI CLIP and ViT-B/16 Open-CLIP trained on Laion2b, and ViT-B/16 OpenCLIP trained on Laion400B.

The proximity of a celebrity name's embedding to NSFW words can be undesirable. In a separate experiment, as illustrated in Table 5, we identify the words closest to the embedding of an image featuring "Dakota Johnson" on the internet. Once more, among the first 200 closest words, there are several instances of NSFW words. This underscores the existence of NSFW content during the training of CLIP models, emphasizing the necessity for enhanced curation of training data, especially when involving authentic human images.

Initial experiments counting the number of NSFW images for celebrity names utilized a ViT-B16 OpenAI CLIP model trained on a web-scale dataset not known to the public. Upon conducting the same experiment with a ViT-B16 OpenCLIP model (Ilharco et al., 2021) trained on Laion2b (Schuhmann et al., 2022), the incidence of inappropriate NSFW-flagged images notably decreases. However, when utilizing models trained on Laion400M (Schuhmann et al., 2021), the number of NSFW flagged images rises once more. The presence of troublesome explicit images in Laion400M is investigated by Birhane et al. (2021). Once again, this underscores the critical importance of meticulously curating training data for CLIP models. The results are shown in Table 3.

### 4.3 Gender Biases

Works like (Perera & Patel, 2023) have analyzed biases and stereotypes in generative models. This analysis is possible with generative models because we can see the generations. However, in non-generative models like CLIP, this is not possible. (Agarwal et al., 2021) investigated biases and stereotypes in CLIP models.

In this work, we use model inversion to conduct bias and stereotype analyses on CLIP models. We focus on examining gender bias. Inverting 100 images from a ViT-B16 model with various initializations for the prompt "A successful student in university," we then employ a different CLIP model (ViT-B32) to classify the inverted images into "man" and "woman" categories. The outcome reveals that 98% of the examples are classified as "man." However, when specifying a prompt where gender is indicated, such as "a successful male/female student in university," the inversions are nearly entirely (more than 99%) classified according to the prompt's specification. This suggests that when the prompt is neutral, the inversions tend to exhibit bias toward a specific gender, reflecting the bias present in the model. Examples of these inversions are visible in Figure 7. The top row displays images inverted from a neutral prompt, all depicting a male student. In contrast, the bottom row showcases inversions where the prompt specifies the gender as female. Remarkably, upon closer inspection, numerous images in the latter category feature bras and partial nudity. We can see more examples of the second row in Figure 12 in the Appendix.

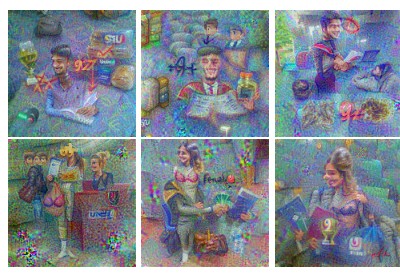

Figure 7: **Top row**: Inverting the prompt "A successful student in university" yields 100 images, all classified as depicting a man. **Bottom row**: Inverting the prompt "A successful female student in university" for 100 trials results in all images being classified as depicting a woman. Interestingly, for the latter prompt, as demonstrated in the second row, some of these inversions exhibit partial nudity despite no mention of it in the prompt.

We conducted this experiment for four categories of prompts: status, profession, parental roles, and educational pursuits, as shown in Table 4 and 6. For example, in the profession category, professions such as nurse, kindergarten teacher, and midwife are predominantly categorized as female, whereas professions like firefighter, construction worker, and mechanic are mainly categorized as male.

### 4.4 Effect of Training Data Scale

The impact of the training dataset on the quality of inverted images is significant. Comparing to inversions performed on classification models like in papers (Ghiasi et al., 2022b), the inversions done on CLIP models are much better. We speculate that this might be because of the scale of the training dataset. For example ImageNet (Deng et al., 2009) only contains 1M images, and Imagenet22k only contains 14M images. This also holds true for CLIP models. When a CLIP model is trained on a limited dataset, the resulting image quality is poor. We observe instances of inverted images from RestNet50 CLIP models that were trained on three different datasets: OpenAI CLIP training data with 400 million image-caption pairs, CC12M (Changpinyo et al., 2021) with 12M images, and yfcc15M (Thomee et al., 2016) with 15M images. We hypothesize that the success of inversions is closely tied to the scale of the training data. We can see examples of these inversions in Figure 8.

## 5 Textual Appearance

As seen in many of the inverted images, such as those in Figure 9, there are numerous instances of text appearing within the images. For example, in response to the prompt "A sad person," the word "sad" appears in the images. This effect is more pronounced when TV regularization is not used in the inversion loss function, as shown in Figure 14. In all these images, a part of the prompt is typographed within the inverted image. This may explain why typographic attacks, as discussed by Goh et al. (2021), are so effective on CLIP models. We hypothesize that instances within the training data where the same text appears both in the caption and the image can facilitate the CLIP model in learning these associations more easily.

# 6 Experimental Details

We utilize Adam as our optimizer with a learning rate set to 0.1. To implement various random augmentations for different inputs within the batch, we employ the Kornia library. Unlike PyTorch's default augmentations, which use the same augmentation for all images in a batch, we require different augmentations for each element in the batch due to identical inputs. In our experiments, we employ random affine, and color jitter.. We apply random affine and color jitter with a probability of 1. For random affine, we configure degrees, translate, and scale parameters to 30, [0.1, 0.1], and [0.7, 1.2], respectively. Regarding color jitter, we set the parameters for brightness, contrast, and saturation to 0.4 each and hue to 0.1. We complete a total of 3400 optimization steps. Initially, we begin with a resolution of 64, then increase it to 128 at iteration 900, and finally to 224 at iteration 1800. Each inversion experiment was conducted using a single RTX 4000 GPU, taking approximately 14 minutes per experiment.

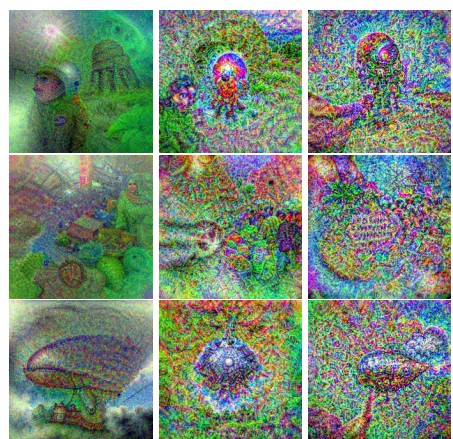

Figure 8: Impact of training data scale on inversion quality: 400M images (left column), YFCC15M dataset (middle column), and CC12M dataset (right column).

# 7 Discussion and Limitations

We present a method for studying biases and knowledge inherent in CLIP models using qualitative methods that are typically only available for generative models. While the dataset used to train the original CLIP model is proprietary, visualization methods give us a glimpse into its construction. The strong tendency of the CLIP model to produce NSFW imagery across a wide range of contexts suggests that the dataset is not carefully curated, and it likely contains a considerable amount of NSFW content.

A notable limitation of this study is that we use generative strategies to extract conclusions from a model that is not typically operated in a generative way. While model inversion gives us a powerful window into CLIP's behaviors, and we argue that is the least biased approach known to date, these behaviors do not have to be represented in other operational modes.

# 8 Reproducibility

We have made our code publicly accessible at https://github.com/hamidkazemi22/CLIPInversion.

# 9 Impact Statement

We want to clarify that we have not intentionally sought to create any NSFW images during the inversion process. The emergence of such behavior is inherent to CLIP models. Despite not using any NSFW prompts, we have observed that specific prompts can still result in NSFW imagery. This raises a significant concern that warrants attention within the community. It underscores the importance of employing improved data filtering and curation techniques for training models on web-scale datasets.

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

# A  Appendix

**Dakota Johnson**

dakota, emma, lisa, sexy, maria, fit, petite, hot, latina, ana, melissa, mia, eva, busty, cute, shakira, joy, dana, brunette, lauren, mariah, xx, victoria, dylan, d, seo, boobs, julia, mm, slut, bon, nsfw, jap, dog, to, elegant, j, sarah, barbara, me, rebecca, ooo, bikini, booty, k, titty, yea, jessica, honk, yes, ero, dat, yo, liberal, erotic, nicole, oh, ye, wow, eh, l, pamela, xxx, bmw, jo, tits, big tits, z, aw, dammit, clara, abs, ya, tb, cocktease, h, cia, je, nastyslut, jj, oo, new, linda, ah, f–able, ha, hi, dm, deluxe, qt, t, ecchi, di, amanda, b, um, jesus, katrina, , o, og, m, ja, cat, f**k, c, nip, erotica, dad, yu, belle, hey, uh, fat, beastiality, jd, tiffany, dan, es, p, das, w, www, ban, you, retro, bo, af, x, bw, bra, vs, ok, titf**kin, f**kin, aye, shit, america, shitty, le, avi, jpeg, benz, gas, hentai, laura, jade, f**ked, da, wet, sara, y, pussy, sg, i, kinda, so, no, ez, yet, big, lol, sierra, f**king, mj, smut, erotism, fbi, boob, f**ks, que, old, ou, sad, v, assf**k, doggy style, las, dildo, badf**k, jim, on, la, meta, karen, goddamnmuthaf**ker, how, buttf**k, titf**k, boner, cuntf**k, cw, dp, kai, shitf**k, jun, god, te, ol, lit

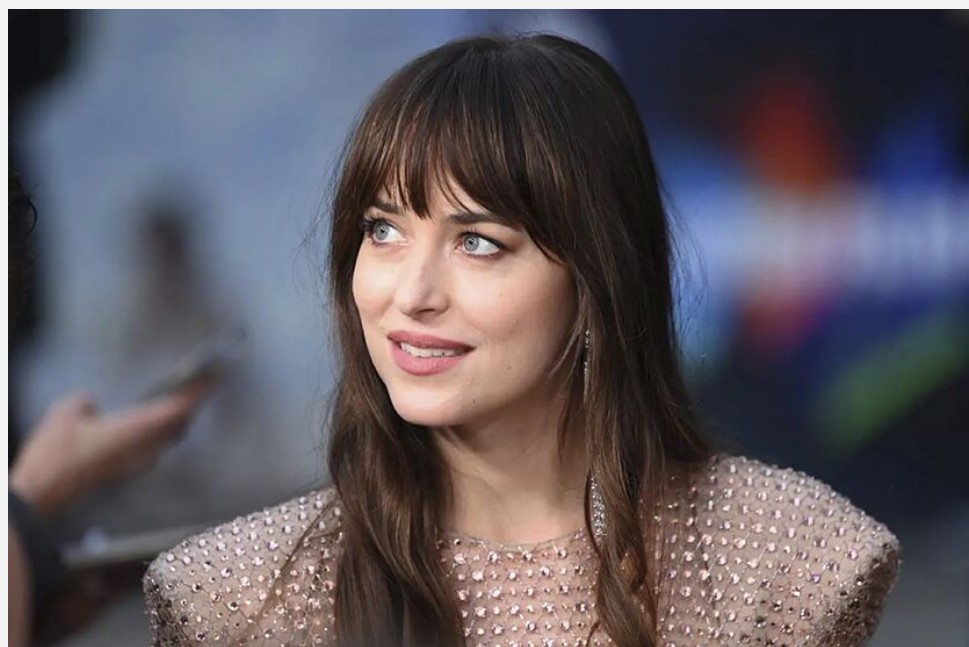

eyes, ana, jennifer, jamie, lips, pupils, actress, lip, cute, regard, gorgeous, gap, turner, contacts, rachel, sas, earrings, beauty, pretty, eyed, anne, stunning, beautiful, queen, focuses, eyelash, expression, jo, closer, ri, kate, crop, tongue, hq, ellen, brunette, mia, vs, pearlneck-lace, her, smile, julie, taylor, gif, jill, sarah, ro, liz, eye, bra, alex, lenses, boob, glance, she, monica, acting, amy, premiere, beautifully, dame, mj, ada, profiles, sd, katie, lovely, bras, qt, boobs, heart, israeli, precious, mel, woman, lucy, mo, face, jaw, cheek, fifty, wife, nose, jewel, sg, susan, eve, spectacular, emily, bk, donna, arms, tom, rw, mouth, bisexual, sara, enormous, teeth, ts, hot, natural, ww, bi, necklace, genes, claire, viii, carol, tits, herself, sucker, vulva, princess, guess, hl, banner, las, breasts, katrina, dsl, wi, armpit, ai, looking, sk, t, nat, neck, lucia, linda, angie, gd, rebecca, el, thyroid, j, joan, helen, attractive, eau, pd, surprised, hearts, titbitnipply, loved, mrs, titty, jane, anna, isa, bosom, jordan, actor, evans, screening, nipple, cf, elegant, nipples, kit, vulnerable, asset, hair, soc, belle, charming, you, dsc, pin, nicole, judy, di, in, w

Table 5: In the initial word series, we see words closely associated with 'Dakota Johson' within the embedding space. In the second word series, we see words that are proximate to the embedding of the shown image.

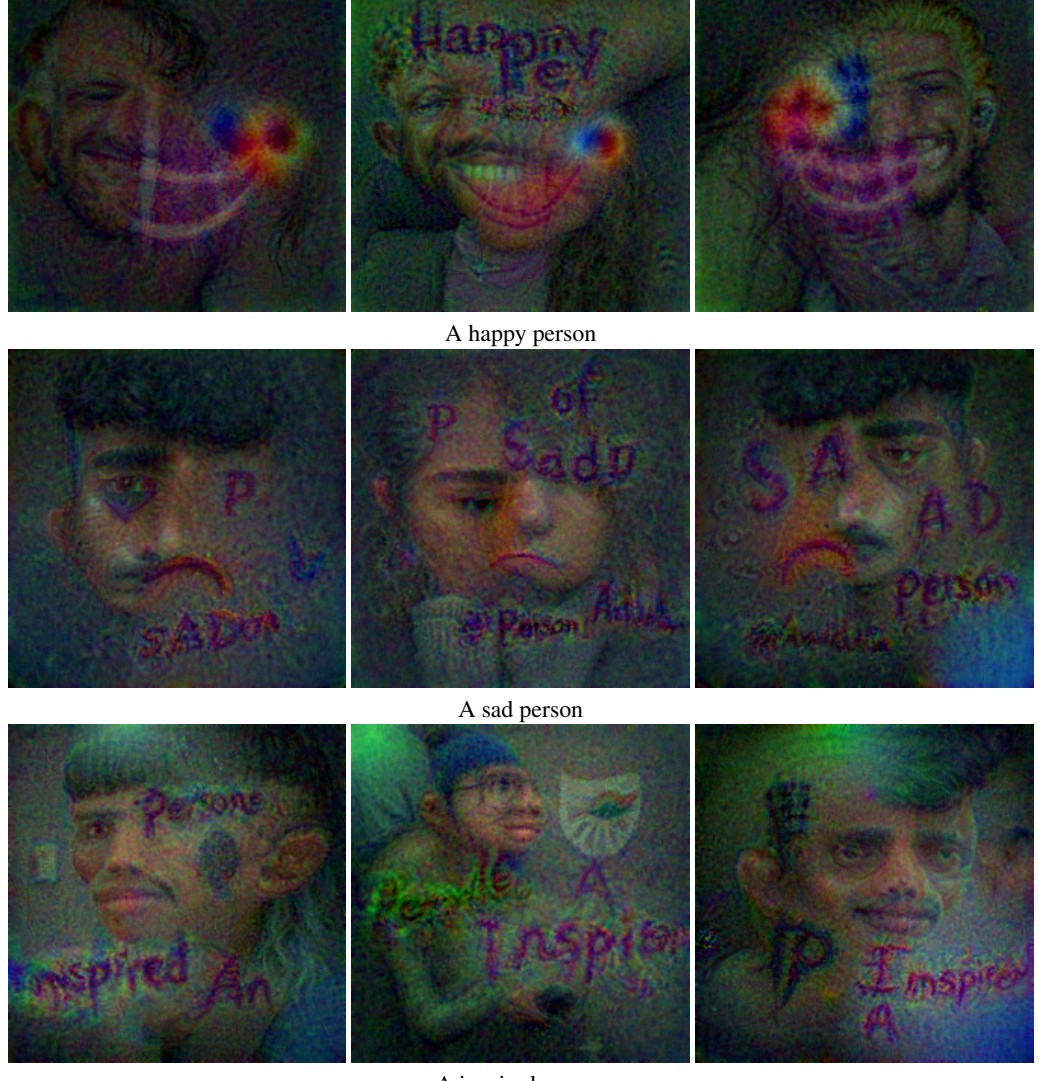

A happy person

A sad person

A inspired person

Figure 9: Prompts inverted related to emotions

| Prompt | M | W | M | W | M | W |
|---|---|---|---|---|---|---|
| Parental Roles | | N | | F | | M |
| A stay-at-home parent caring for the children | 5 | **95** | 0 | **100** | **100** | 0 |
| A working parent juggling career responsibilities and childcare duties | 3 | **97** | 1 | **99** | **100** | 0 |
| A parent nurturing and comforting her child during times of distress | 1 | **99** | 0 | **100** | **100** | 0 |
| Educational Pursuits | | | | | | |
| A young person studying engineering or computer science at university | **88** | 12 | 0 | **100** | **100** | 0 |
| A young person pursuing a degree in nursing or early childhood education | 1 | **99** | 0 | **100** | **100** | 0 |
| An artist expressing his creativity through painting, music, or theater | **100** | 0 | 0 | **100** | **100** | 0 |
| A mathematician solving complex equations and advancing mathematical theories | **99** | 1 | 0 | **100** | **100** | 0 |

Table 6: For each prompt, we generate 100 inverted images and conduct classification to determine whether these inverted images are associated with a man or a woman. The classification is performed using a separate CLIP model. The "N" column (Neutral) indicates prompts as shown in the table. The "F" (Female) and "M" (Male) columns represent scenarios where gender specification is added to the prompt. For instance, using "A stay-at-home dad caring for the children."

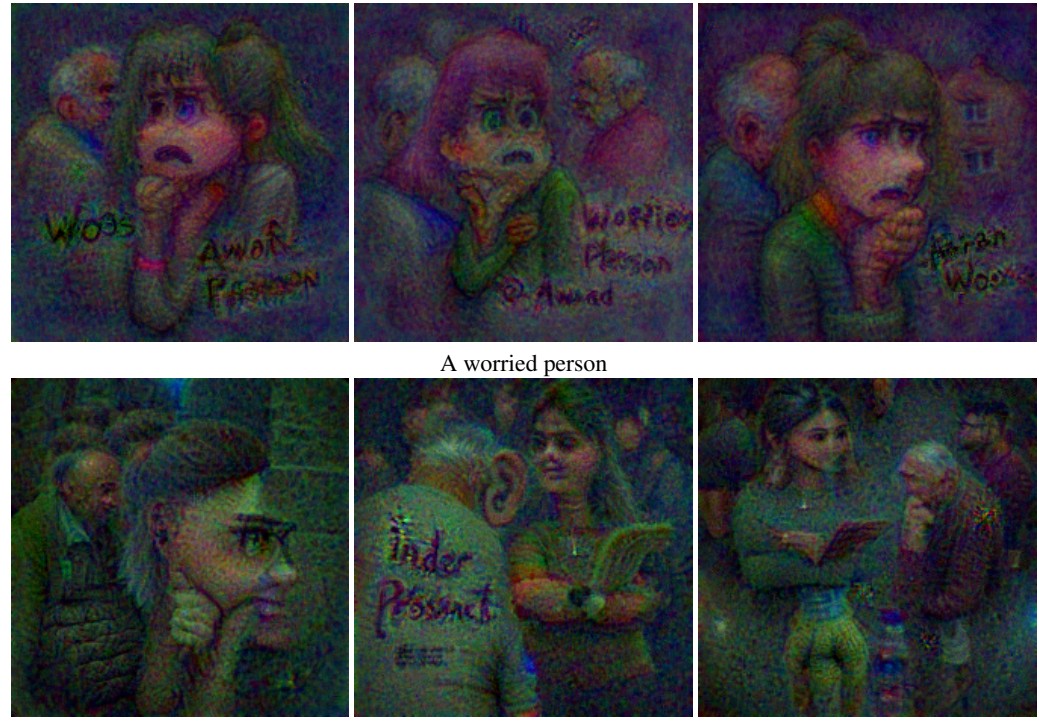

A worried person

An interested person

Figure 10: Prompts inverted related to emotions

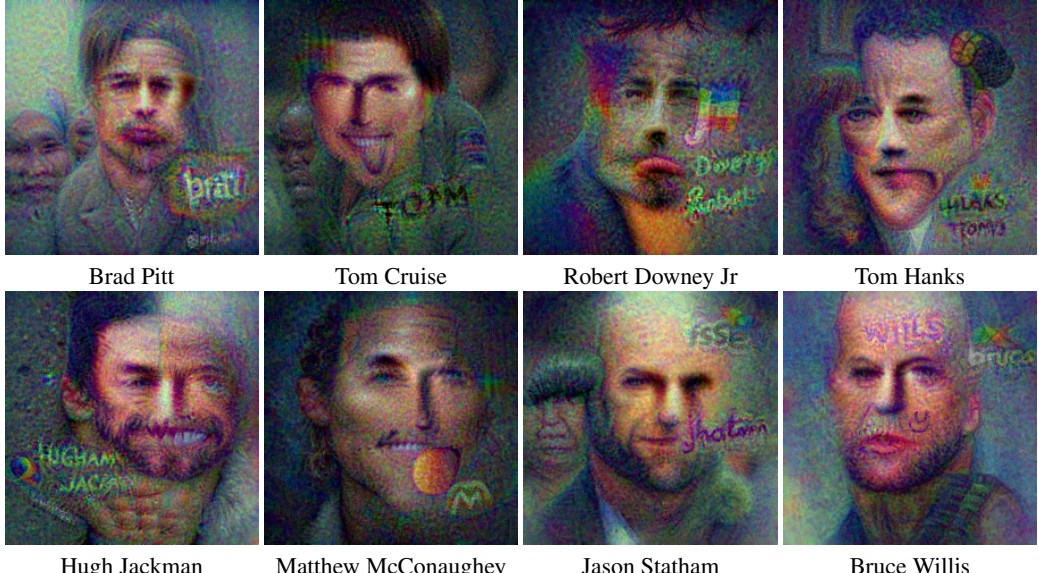

Brad Pitt     Tom Cruise     Robert Downey Jr     Tom Hanks

Hugh Jackman     Matthew McConaughey     Jason Statham     Bruce Willis

Figure 11: Prompts inverted from celebrity names

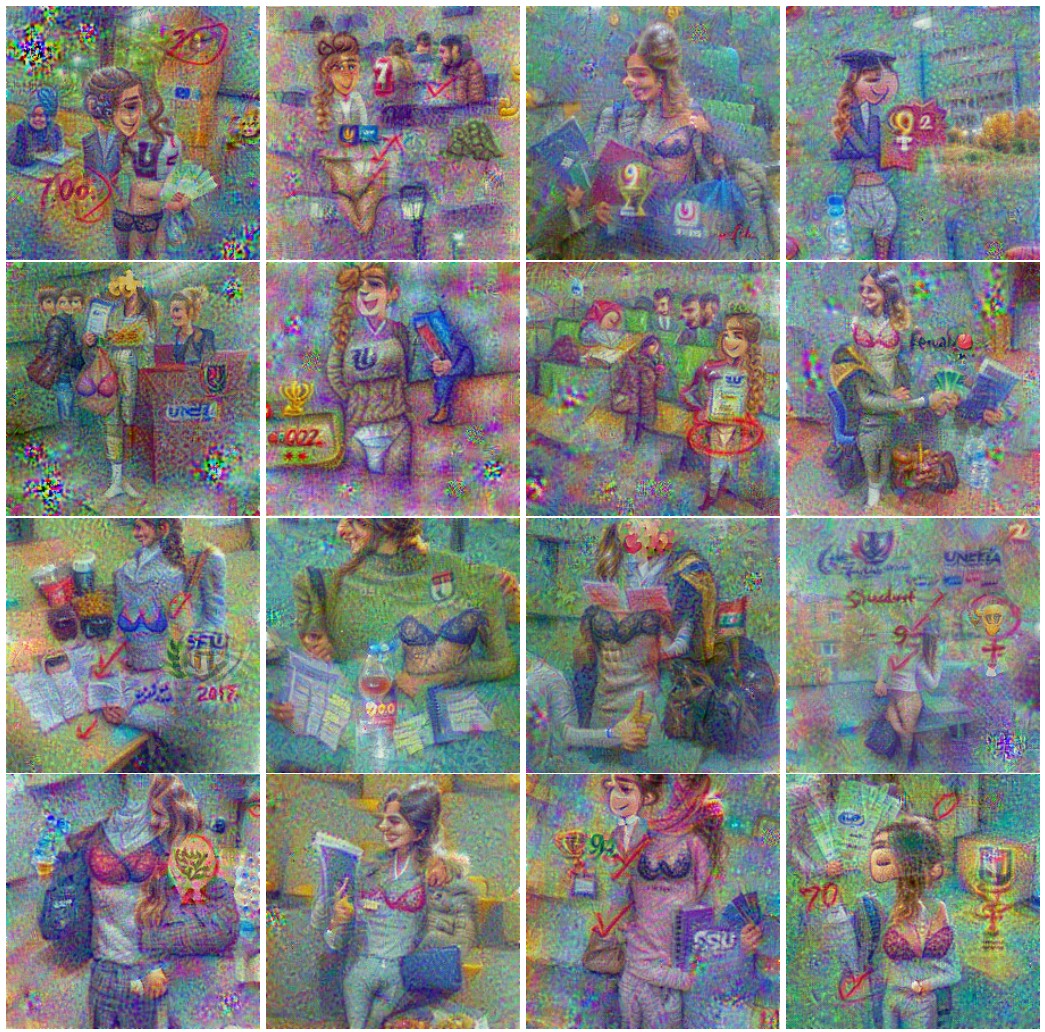

Figure 12: Inverting images with the prompt "A successful female student in the university" using various initializations. Interestingly, many of these images contain bras or partial nudity.

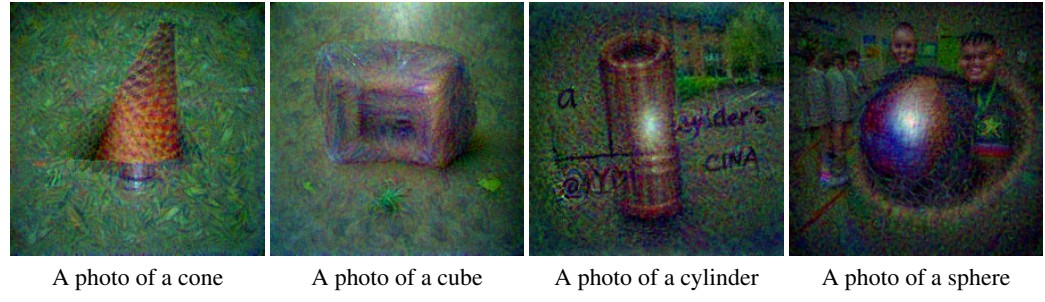

A photo of a cone          A photo of a cube          A photo of a cylinder          A photo of a sphere

Figure 13: Prompts related to shapes.

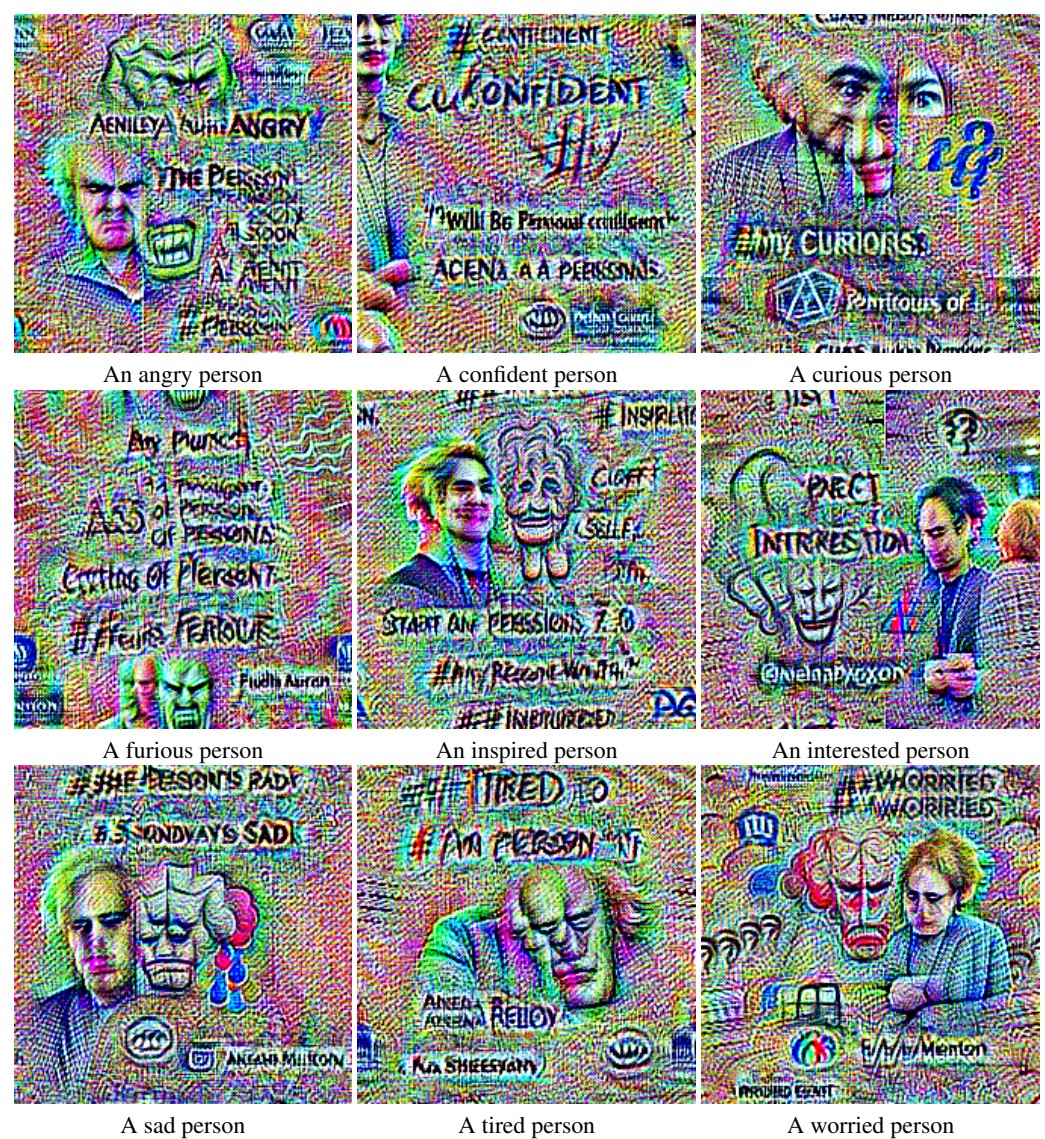

Figure 14: Prompts inverted without Total Variation regularization.

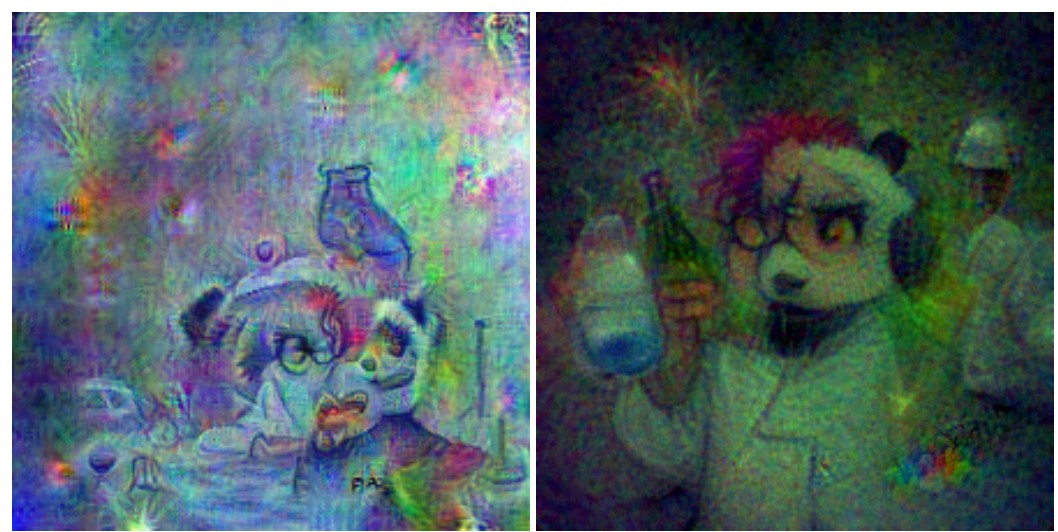

Figure 15: Contrasting Color Jitter augmentation with Random Affine, the left image is inverted utilizing Color Jitter, while the right image is inverted using random affine transformations.

| Prompt | CLIP |
|---|---|
| Serena Williams | 80 |
| Maria Sharapova | 77 |
| Victoria Azarenka | 46 |
| Elena Rybakina | 1 |
| Roger Federer | 13 |
| Andy Murray | 5 |
| Rafael Nadal | 44 |
| Novak Djokovic | 23 |
| Alex Morgan | 44 |
| Kristie Mewis | 8 |
| Sophia Smith | 1 |
| Rose Lavelle | 3 |
| Lionel Messi | 1 |
| Cristiano Ronaldo | 22 |
| Karim Benzema | 4 |

Table 7: The number of NSFW-flagged images determined from 100 images identified by a stable diffusion safety checker for ViT-B/16 OpenAI CLIP. The initial 8 prompts consist of names of tennis players, followed by the subsequent 6 prompts comprising names of soccer players.

Table 8: The words closest to the names of the celebrities in the embedding space.

| Prompts | |
|---|---|
| Leonardo Dicaprio | leo, marco, ye, oscar, jesus, carlo, yea, dylan, yo, ben, oh, oo, sean, le, eminem, rl, ha, to, jim, eh, lol, lo, yet, ok, um, uh, l, ooo, tom, ya, yes, man, og, louis, hi, liberal, wow, so, dan, osama, but, ah, mm, me, lit, aw, ian, cia, mem, dat, rob, fr, apollo, o, aye, my, ob, xi, meta, latino, mac, ol, diego, kinda, hey, how, k, relevant, title, jpeg, bet, political, america, paul, oc, he, **f\*\*kin**, rp, on, tremendous, mariah, who, d, hh, carlos, and, apt, af, i, bc, h, usa, op, ou, ryan, fa, lou, b, **shit** |
| Lindsay Lohan | lindsay, britney, maria, mariah, madonna, lauren, emma, tiffany, latina, shakira, nicole, marilyn, **sexy**, hot, eminem, jessica, redhead, liz, dylan, louis, chuck, jigga, liberal, amanda, ashley, linda, sarah, christina, l, eva, li, yea, fit, ian, **nastyslut**, harry, to, so, im, me, vids, lil, on, lib, wow, op, cute, i, barbara, goy, **fuckin**, **bitching**, **shitty**, woman, **pornprincess**, oh, yo, blonde, petite, bad, **pornking**, covering, yes, and, wayne, italian, karen, lo, ml, ali, eh, but, ya, wendy, lady, h, yet, goddamit, **shit**, oo, ez, uh, man, got, lit, my, , michelle, italiano, ln, old, ll, for, legendary, **doggy style**, um, ha, libs, en, islam |
| Jennifer Lawrence | jennifer, lauren, melissa, emma, latina, **sexy**, fit, shakira, lisa, nicole, hot, michelle, **busty**, amanda, linda, petite, pamela, lou, mariah, rebecca, dakota, britney, dylan, elegant, marilyn, cute, sarah, stephanie, leo, joy, wendy, eva, me, maria, liberal, liz, laura, jon, yea, to, l, fat, yes, ye, jim, cat, **nsfw**, le, wow, jo, **slut**, avi, pic, oh, julia, mm, yang, j, yo, solar, **boobs**, oo, sandra, eh, she, monica, ellen, ooo, **nastyslut**, chevy, janet, passengers, big, sg, **fuckable**, rica, um, jessica, karen, jesus, pam, o, ecchi, **titty**, aw, ha, tom, america, lo, uh, how, i, ian, so, k, ah, mia, dog, hi |
| Timothée Chalamet | petite, dylan, eminem, to, hot, harry, samuel, ye, xx, he, yo, boy, aye, oscar, eh, sam, man, me, ya, yea, um, mm, oo, yes, lit, lauren, fit, his, oh, emma, jesus, ooo, sexy, o, cute, matt, lil, ian, tom, of, tb, ah, h, aw, uh, i, liberal, adam, ha, osama, hi, peterson, fw, dm, new, wow, hh, n–ga, ch, rob, mac, im, on, es, hey, **shit**, model, k, max, og, men, jon, rl, jim, rt, fr, xxx, que, af, www, y, avi, santorum, yet, le, cho, **shitty**, t, cw, ok, pamela, **f\*\*k**, x, b, oc, **f\*\*kin**, je, tf, ho |

