# OpenReview forum: "What do we learn from inverting CLIP models?"
_NeurIPS.cc/2024/Workshop/SafeGenAi — SafeGenAi Poster_

### Official Review · Reviewer_sgY2 · 2024-10-08
**Interesting CLIP bias analysis, but lacks polish and systematic evaluation.**

**Rating:** 6
**Confidence:** 5

**Review:**

Strengths:

This work presents an interesting analysis of the popular CLIP foundation models by using standard inversion techniques. It is the first work to interrogate CLIP with these methods, uncovering proof of significant biases in the model.

The paper is generally easy to follow, with sufficient background to motivate why looking into CLIP-like models is important. The methodology for generating the inversions seems fine, looking similar to other vision encoder inversion techniques.

The results are intriguing and make sense.


Weaknesses:

The paper's presentation is quite sloppy:
- There's a lot of unpolished text " By utilizing the extensive set of prompts available for inverting CLIP models, we delve into analyzing various aspects of this family of models." for example, adds nothing to the paragraph.
- Missing citations "Concerns raised by (?)"
- vspace issues make it very hard to read (e.g., end of page 6 / start of 7)
- table 4 is too wide
- figures and tables are not formatted for a 1 column paper and is jarring
- raw presentation of extensive profanity
- section 5 appears suddenly, with little context and there is no conclusion/discussion/ethical implication/broader impact/etc.
- far too many references to the appendix. Many subsections do not make sense without references the supplemental material.

The experimental design is a bit too qualitative in my opinion. While I don't mind missing ablation experiments (on different models or different optimization setups), the primary experimental setup appears to be arbitrary: random celebrities, random phrases " A beautiful landscape", random professions. The lack of systematic study inclines me to believe this work does not generalize.

Another example is the experiment is this experiment: "we utilize a word list including 10,000 most common English words1, Naughty, Obscene, and Otherwise Bad Words2, Names for body parts 3, Offensive/Profane Word List 4, 11913 words in total". You have 10k low resolution words, and 1913 high resolution nouns/adjectives. It's not surprising to me the list of bad words is more relevant simply due to random choice of highly descriptive words on these specific prompts. Also, it's somewhat unusual to do single/dual word embeddings only in the text-space as the model is trained on full sentences.

Table 4 is suspect too, as using another CLIP model to be a Man/Woman classifier is self-reinforcing bias. That experiment does not indicate in which CLIP model the bias lies. Furthermore, the model was never trained on inverted images, so I have little confidence in the generalization of bias from this experiment.

Overall, I think this work is close to being a valuable exploration of CLIP bias, but more work is needed. I could see a full conference version of this paper where more metrics are introduced, a more systematic way of selecting sentences is used, ablations on the optimization procedure, and experiments on other modality CLIP-like models (like CLAP).

All that said, in my mind, the main question is would this work be of interest to workshop attendees: the answer is clearly yes.

---

### Official Review · Reviewer_xfPP · 2024-10-11
**Comments**

**Rating:** 6
**Confidence:** 2

**Review:**

This paper applies an inversion-based approach to examine CLIP models, shedding light on their semantic capabilities and biases. While the approach itself is not novel, the paper provides valuable insights into CLIP's behavior and its potential for unintended outputs, particularly in relation to concept blending and gender bias. Several strengths of this work deserve recognition:

Insightful Observations: The findings regarding CLIP's ability to blend concepts and the appearance of NSFW images during inversion, even for seemingly harmless prompts, are important contributions. These observations highlight critical issues for practitioners and researchers working with CLIP models, especially when deploying them in sensitive or public-facing applications.

Addressing Biases: The paper makes a notable contribution by emphasizing the presence of gender biases in the inverted images. This aligns with growing concerns in AI regarding fairness and ethical AI deployment. The use of inversion to expose these biases provides a tangible way to analyze model behavior that might otherwise remain hidden.

Relevance to Model Interpretability: The study of CLIP models through inversion offers a practical method for understanding how these models internally represent and manipulate concepts. This could be valuable for improving transparency in vision-language models and informing the development of fairer, more robust models.

However, there are certain aspects of the paper that could be further improved:

Lack of Novelty in Methodology: While the application of inversion to CLIP models is interesting, the inversion-based approach itself is not new. The paper could benefit from a more detailed discussion on how this method extends or builds upon previous work in model inversion. Highlighting any unique contributions or adaptations of the inversion technique would strengthen the innovation aspect of the paper.